# *Lactiplantibacillus plantarum* HY7718 Attenuates Renal Injury in an Adenine-Induced Chronic Kidney Disease Mouse Model via Inhibition of Inflammation and Apoptosis

**DOI:** 10.3390/ijms262010052

**Published:** 2025-10-15

**Authors:** Hyeonji Kim, Ji-Woong Jeong, Haeryn Jeong, Daehyeop Lee, Hyeonjun Gwon, Kippuem Lee, Joo-Yun Kim, Jae-Jung Shim, Jae-Hwan Lee

**Affiliations:** R&BD Center, hy Co., Ltd., 22 Giheungdanji-ro 24 Beon-gil, Giheung-gu, Yongin-si 17086, Republic of Korea; skyatk94@gmail.com (H.K.); woongshow@hy.co.kr (J.-W.J.); haeryn@hy.co.kr (H.J.); flywhy7@hy.co.kr (D.L.); hjgwon@hy.co.kr (H.G.); 10002903@hy.co.kr (K.L.); jjshim@hy.co.kr (J.-J.S.); jaehwan@hy.co.kr (J.-H.L.)

**Keywords:** *Lactiplantibacillus plantarum*, chronic kidney disease, kidney function, inflammation, apoptosis

## Abstract

Chronic kidney disease (CKD) causes a variety of health problems including renal dysfunction and cardiovascular disease. This study aimed to investigate whether the probiotic strain *Lactiplantibacillus plantarum* HY7718 (HY7718) can protect against CKD using HK2 cells and a CKD mouse model, generated by feeding mice a diet containing 0.15% adenine. In vitro tests showed that HY7718 was anti-inflammatory in H_2_O_2_-treated HK2 cells and reduced apoptosis of tumor necrosis factor-α/cycloheximide-induced HK2 cells. In the adenine-induced CKD model, markers of renal dysfunction (blood urea nitrogen (BUN) and creatinine (Crea)) and inorganic calcium and phosphorus were markedly increased. However, oral administration of HY7718 (10^8^ colony-forming units/kg/day) significantly attenuated these increases. HY7718 also reduced the kidney histopathological score, including tubular necrosis, cast formation, and tubular dilatation, as well as the mononuclear cell infiltration score in kidney tissue, suggesting that it could reverse the progression of CKD. Additionally, HY7718 downregulated the renal expression of pro-inflammatory cytokine genes and members of the TLR/NF-κB signaling pathway. Furthermore, HY7718 reduced tubule apoptotic cells and expression of apoptosis-related genes, indicating that it is potentially renoprotective. These results demonstrate that supplementation with the probiotic HY7718 can ameliorate CKD symptoms by improving renal function and reducing kidney injury.

## 1. Introduction

Chronic kidney disease (CKD) is a progressive and irreversible condition marked by the gradual loss of renal function [1]. CKD has emerged as a major public health concern, affecting approximately 10–15% of the global population, with increasing prevalence driven by aging demographics and the rising incidence of diabetes and hypertension [2,3]. The clinical features of CKD include not only renal dysfunction and increased proteinuria, but also various complications, including electrolyte imbalance, cardiovascular disease, bone disease, and decreased immune function [4,5]. CKD is classified into five stages based on the level of kidney function, primarily determined by the estimated glomerular filtration rate (eGFR). These stages range from mild kidney damage with normal or increased GFR (Stage 1) to kidney failure, or end-stage kidney disease (ESKD) (Stage 5) [4,6]. Current therapeutic approaches include medications that treat various aspects of the disease and its complications, including renin-angiotensin system inhibitors and sodium-glucose cotransporter (SGLT2) inhibitors [7,8]. In addition, the progression of CKD to ESKD inevitably requires kidney replacement therapies, such as dialysis or transplantation [9,10]. However, these therapies often have only limited benefits and may lead to undesirable side effects including decline in quality of life and high risk of premature death, underscoring the need for alternative strategies to improve patient outcomes [11,12].

Probiotics are defined as “Live microorganisms that, when administrated in adequate amounts, confer a health benefit on the host” in 2001 [13]. *Lactiplantibacillus plantarum*, a widely used probiotic strain, is a Gram-positive, nonmotile, non-sporeforming, and bacilli-shaped bacterium [14]. It is a versatile microorganism found in a wide range of ecological niches ranging from fermented food to the gastrointestinal tract. Microorganisms belonging to the genus *Lactiplantibacillus* demonstrate acid and bile tolerance, adherence to intestinal epithelium, and immunomodulatory properties [15]. Several strains of *L. plantarum* are reported to have beneficial effects, such as maintaining gut flora balance, reinforcing intestinal barrier function, protecting skin health, counteracting liver diseases, and improving mental health [16,17,18]. In addition, recent studies have reported that *L. plantarum* can improve clinical symptoms of CKD in a rodent model by reducing inflammation, protecting kidney function, and modulating gut microbiota [10,19,20].

Various animal models have been used to elucidate the pathophysiological mechanisms of CKD. The adenine-induced CKD model is one of the most commonly used [21]. Adenine, a purine nucleotide base, has a central role in various biological processes; for instance, it is a component of DNA and RNA [22]. Adenine at physiological levels is metabolized by adenine phosphoribosyl transferase to adenosine monophosphate (AMP), which is then further metabolized to adenosine, inosine, hypoxanthine, and xanthine. These compounds are converted to uric acid, and excreted through the kidney [23]. However, high doses of adenine cause the formation of 2,8-dihydroxyadenine (2,8-DHA), which accumulates in the kidneys and leads to kidney damage, including loss of kidney function, increased blood urea nitrogen (BUN) and creatinine (Crea) levels, and renal tubular injury [21,24,25]. Additionally, 2,8-dihydroxyadenine crystals cause renal tubular occlusion and peritubular capillary damage, leading to CKD, which is associated with a multifactorial pathophysiology, including renal inflammation and programmed cell death [21,26].

In our previous study, we found that *Lactiplantibacillus plantarum* HY7718 (HY7718), isolated from Korean traditional fermented foods, has production, storage, and probiotic properties that make it suitable for industrial applications. Additionally, *L. plantarum* HY7718 has been reported to downregulate the secretion of pro-inflammatory cytokines and production of reactive oxygen species (ROS) in vitro [27]. Furthermore, HY7718 exerts anti-inflammatory effects by suppressing inflammation marker levels in a colitis mouse model [28]. However, despite evidence that HY7718 has anti-inflammatory properties, no studies to our knowledge have examined whether *L. plantarum* HY7718 ameliorates CKD.

The purpose of this study is to determine whether the probiotic strain *L. plantarum* HY7718 alleviates CKD. We evaluated the anti-inflammatory and anti-apoptosis effects of HY7718 in proximal tubular HK2 cell lines and in an adenine-induced CKD mouse model. In parallel, serum biochemical and histopathological analyses were performed to investigate whether HY7718 is renoprotective in the CKD model.

## 2. Results

### 2.1. Effect of HY7718 on Viaibility of H_2_O_2_-Treated HK2 Cells

We used H_2_O_2_-treated HK2 cells to investigate the effects of HY7718 on cell survival rates and lactate dehydrogenase (LDH) levels. In the 3-(4,5-dimethylthiazol-2-yl)-2,5-diphenyltetrazolium bromide (MTT) assay, the cell viability of H_2_O_2_-treated HK2 cells was significantly decreased to 64.02% compared with that of untreated cells (*p* < 0.001) (Figure 1A). Pre-treatment with 10^6^ and 10^7^ colony-forming units (CFU)/mL of HY7718 raised the survival rates to 73.22% and 74.23%, respectively (*p* < 0.001). In Figure 1B, LDH assays showed that LDH release levels were significantly higher (at 136.5%) in H_2_O_2_-treated cells than those in H_2_O_2_-untreated cells (100%; *p* < 0.001). HY7718 treatment reduced LDH levels concentration-dependently (10^6^ CFU/mL, 129%; 10^7^ CFU/mL, 120.9%), but the difference was only significant at 10^7^ CFU/mL (*p* < 0.01).

### 2.2. Measurement of Pro-Inflammatory Cytokine Levels in H_2_O_2_-Treated HK2 Cells

We next examined whether HY7718 affects pro-inflammatory cytokine levels in H_2_O_2_-treated HK2 cells using gene expression and ELISA assays (Figure 2). After inducing oxidative stress with H_2_O_2_, the expression levels of *TNF* and *IL-6* increased significantly by 7.81-fold and 1.56-fold, respectively, compared with those of H_2_O_2_-untreated cells (*p* < 0.001). At 10^6^ CFU/mL, HY7718 pre-treatment decreased TNF and IL-6 mRNA expression to 4.93-fold (*p* < 0.001) and 1.41-fold, respectively. At 10^7^ CFU/mL, HY7718 significantly decreased *TNF* and *IL-6* mRNA expression to 4.24-fold (*p* < 0.001) and 1.27-fold (*p* < 0.01), respectively.

ELISA data of pro-inflammatory cytokine levels showed a very similar pattern to that of the gene expression data. H_2_O_2_ treatment significantly increased TNF (to 176.2%) and IL-6 (to 172.2%) levels in HK2 cells with respect to those in H_2_O_2_-untreated cells (*p* < 0.01). However, pre-treatment with 10^6^ and 10^7^ CFU/mL of HY7718 significantly lowered TNF levels to 100.3% (*p* < 0.01) and 92.1% (*p* < 0.001), respectively, while those of IL-6 were lowered to 128.0% at 10^6^ CFU/mL and 117.6% at 10^7^ CFU/mL, but only treatment with 10^7^ CFU/mL HY7718 resulted in a significant difference (*p* < 0.05).

All measured indicators demonstrated a concentration-dependent reduction in pro-inflammatory cytokines. Taken together, these results show that HY7718 suppresses pro-inflammatory cytokine gene expression and secretion in oxidative stress-induced renal cells.

### 2.3. Effect of HY7718 on Cell Viability of TNF+CHX-Treated HK2 Cells

To evaluate whether HY7718 attenuates apoptosis in renal cells, an apoptosis model was established by treating HK-2 cells with 5 ng/mL tumor necrosis factor-alpha (TNF-α) and 25 nM cycloheximide (CHX). Figure 3A shows that the marked reduction in cell viability induced by TNF+CHX (*p* < 0.001) was attenuated by pre-treatment with different concentrations of HY7718 (*p* < 0.05).

### 2.4. The Effect of HY7718 on the Apoptosis Marker Caspase-3 in TNF+CHX-Treated HK2 Cells

Next, we assessed whether HY7718 treatment affects caspase-3 activity and cleaved caspase-3 levels in HK2 cell lines. As shown in Figure 3B, relative caspase-3 activity was increased by TNF+CHX treatment (*p* < 0.001) and the increase was significantly reduced by HY7718 treatment. Changes in the levels of cleaved caspase-3 paralleled those in caspase-3 activity. In Figure 3C, the cleaved caspase-3 level in TNF+CHX-treated cells (179.16 ± 11.82 pg/mL, *p* < 0.001) was significantly higher than that in untreated cells (31.53 ± 0.15 pg/mL). Pre-treatment with HY7718 significantly reduced this increase in the cleaved caspase-3 level (141.68 ± 9.53 pg/mL at 10^6^ CFU/mL and 138.72 ± 5.88 pg/mL at 10^7^ CFU/mL, *p* < 0.001).

### 2.5. Effects of HY7718 on Psysiological Indicators in CKD-Induced Mice

To investigate whether HY7718 affects physiological parameters in mice with adenine diet-induced CKD, we measured changes in body weight, water consumption, and food intake in adenine-induced CKD mice with and without HY7718 or allopurinol (APN) treatment (Figure 4A and Appendix A). Adenine-induced CKD mice showed increased body weight loss compared with control mice (*p* < 0.001). APN- and HY7718-treated CKD mice showed less body weight loss than the untreated CKD mice (n.s). Figure 4B,C shows the spleen weights and the spleen to body weight ratios of adenine-induced CKD mice were significantly higher at 134.7% and 173.2%, respectively, than those of control mice. However, APN or HY7718 treatment reduced significantly the increases in a spleen/body weight ratio to 155.9% and 147.1%, respectively, of those of control mice. Figure 4D,E shows that kidney weights and kidney/body weight ratios of adenine-induced CKD mice were higher at 110% (*p* < 0.01) and 133.5% (*p* < 0.05), respectively, than those of control mice. Treatment with HY7718 significantly lowered the increases in these parameters to 99.8% (*p* < 0.01) and 120.1% (*p* < 0.05), respectively. Figure 4F shows that the kidneys of control mice exhibited normal sizes and a reddish-brown coloration, whereas those of adenine-induced mice (CKD mice) exhibited a yellow discoloration. APN or HY7718 treatment restored the reddish-brown coloration.

### 2.6. Effects of HY7718 on Blood Biochemistry in CKD-Induced Mice

Next, we analyzed kidney function parameters in serum samples. Adenine-induced CKD mice showed significant elevations in the serum concentrations of two kidney function markers, BUN and Crea, to 70.73 ± 17.37 mg/dL and 0.94 ± 0.13 mg/dL, respectively, compared with those in control mice (Figure 5A,B). HY7718 treatment of these CKD mice attenuated these increases in BUN (to 54.07 ± 4.40 mg/dL, *p* < 0.05) and Crea (to 0.68 ± 0.16 mg/dL, *p* < 0.01). Figure 5C,D shows that serum levels of calcium (Ca) and phosphorus (P) in CKD mice tended to be significantly higher than those in control mice. HY7718 treatment reduced these levels to below those in untreated CKD mice (Ca; *p* < 0.01, P; not significant). APN treatment also attenuated the increases in serum levels of BUN, Crea, Ca, and P in CKD mice; however, the effect of APN was not as pronounced as that of HY7718.

### 2.7. Effects of HY7718 on Kidney Histopathology

Hematoxylin and eosin (H&E) stained kidney sections showed a normal kidney histology in control mice but an abnormal kidney histology in adenine-induced CKD mice, including tubular necrosis, cast formation, and tubular dilatation (Figure 6A) [29]. HY7718 treatment partially restored the normal kidney histology of these mice. As illustrated in Figure 6B, the kidney histopathological score was significantly higher in adenine-induced CKD mice (8.50 ± 0.55) than in control mice (0.00, *p* < 0.001). HY7718 or APN treatment reduced the kidney histopathological score to 5.67 ± 0.82 (*p* < 0.001) and 6.83 ± 1.33 (*p* < 0.05), respectively (Figure 6C). In addition, adenine-induced CKD mice showed a marked increase in the mononuclear infiltration score (*p* < 0.001), which was significantly attenuated by HY7718 and APN (*p* < 0.05).

### 2.8. Effects of HY7718 on Renal Inflammation-Related Gene Expression

To evaluate whether HY7718 affects the expression of renal inflammation-related genes in adenine-induced CKD mice, we measured the mRNA levels of these genes (*Tnf*, *Il-6*, *Il-1β* and *Ccl2*), including those belonging to the NFκB/TLR signaling pathway (*Tlr4* and *Nfκb1*). Figure 7A–F show that the adenine-induced CKD mice had significantly higher levels of inflammation- and inflammatory signaling pathway-related gene expression than the control mice (*p* < 0.001), and that HY7718 significantly reduced the expression of these genes to levels below those of the adenine-induced CKD mice. APN treatment also downregulated the expression levels of these genes, but significant differences in expression were observed only for *Ccl2* and *Tlr4*.

### 2.9. Effect of HY7718 Apoptosis in Kidney Tissues

Finally, we investigated whether HY7718 affects apoptosis and the expression of apoptosis-related genes such as *Bax/Bcl-2* ratio and *Casp3* in adenine-induced CKD mice. Renal tubule apoptotic cells were detected using the transferase dUTP nick end labeling (TUNEL) assay (Figure 8A) [30]. In the group fed an CKD-induced diet, TUNEL-positive cells (brown color cells) was significantly increased (*p* < 0.001), but these apoptotic cells were reduced by administration of APN and HY7718 (Figure 8B). Figure 8C,D shows that the *Bax/Bcl-2* expression ratio and the *Casp3* expression level in adenine-induced CKD mice were significantly higher at 1.42- and 1.95-fold (*p* < 0.001), respectively, than those in control mice. HY7718 treatment significantly reduced these parameters to 0.93- (*p* < 0.001) and 1.51-fold (*p* < 0.05), respectively. In APN-treated mice, the *Bax/Bcl-2* ratio and expression level of *Casp3* were reduced to 1.29- and 1.77-fold, respectively. These data obviously indicate that the anti-apoptosis effect of HY7718 functions by reducing apoptotic cells and the expression of apoptosis-encoding related genes.

## 3. Discussion

CKD is characterized by gradual and/or progressive damage and loss of function of the kidney that persist for 3 months or more [6]. This study aimed to determine whether the probiotic *Lactiplantibacillus plantarum* HY7718 protects against and ameliorates CKD.

In CKD, oxidative stress refers to an imbalance between harmful ROS and the host’s antioxidant defense system. This causes damage to kidney cells and tissues, accelerating the development and progression of kidney disease [31]. In addition, oxidative stress triggers kidney inflammation, and inflammatory responses can lead to kidney tissue damage that spreads systemically [32]. Pro-inflammatory cytokines, notably TNF-α, IL-6, IL-1β, and CCL2 (MCP-1), promote inflammation and play a significant role in the pathogenesis and progression of CKD [33]. Elevations in the levels of these factors are associated with reduced kidney function, kidney damage, and increased albuminuria in CKD patients [34,35,36].

In this study, we first confirmed that oxidative stress reduces the viability of kidney-derived cells, and that HY7718 protects against this loss of viability. In addition, we found that the overproduction of pro-inflammatory cytokines, such as TNF and IL-6, induced by H_2_O_2_, was significantly attenuated by HY7718 treatment. We also found that HY7718 suppressed the mRNA expression of genes encoding *TNF* and *IL-6*, which was elevated after H_2_O_2_ treatment. Our results suggest that HY7718 may have cell protective and anti-inflammation effects in oxidative stressed renal cells.

Apoptosis (or programmed cell death) is involved in cell loss and progressive damage in CKD by inducing renal tubular cell death and modulating inflammation and fibrosis [37,38]. In other words, apoptosis is closely connected with the pathophysiology of renal diseases. Caspase-3 is a central key enzyme of apoptosis and is a known driver of kidney dysfunction. Activated caspase-3 cleaves key structural proteins, DNA-degrading enzyme (DNase), and cell cycle proteins, ultimately leading to cell death [39,40]. Cleaved caspase-3 is the activated, functional form of caspase-3 [41]. TNF-α and cycloheximide (CHX) together (TNF+CHX) dramatically increases and accelerates apoptosis by inhibiting protein synthesis and removing cellular inhibitors of apoptosis, leading to rapid programmed cell death [42]. In this study, we showed that in TNF+CHX-treated HK2 cell cells, treatment with HY7718 restored cell viability, and decreased caspase-3 activity and the levels of cleaved caspase-3. Thus, HY7718 may prevent renal cellular apoptosis by downregulating caspase-3 activity and the level of cleaved caspase-3 in programmed cell death-induced cells.

To investigate the effect of HY7718 in vivo, C57BL/6 mice were fed a diet containing 0.15% adenine for 3 weeks to induce CKD. The adenine diet resulted in various physiological changes, including increased water intake (Appendix A), loss of body weight, and increases in the spleen or kidney/body weight ratio. The heightened water consumption is a compensatory response to the increased fluid loss caused by adenine-induced kidney damage [43]. Spleen enlargement is also a potential feature of CKD and is caused by immune-related responses or inflammation [44]. The relative kidney weight of adenine-induced CKD mice is significantly increased [45]. APN, used as a positive control in this study, is known to reduce uric acid production by the inhibiting conversion of hypoxanthine to uric acid [46]. In this study, both APN and HY7718 were effective in alleviating the effects of adenine-induced CKD, but HY7718 was more effective than APN in this respect.

Adenine is metabolized to 2,8-DHA, the crystals of which accumulate in renal tubules, leading to increases in blood levels of BUN and Crea, and tubular injury [47,48]. BUN and Crea are waste products that are measured in blood to evaluate kidney function. Abnormally high concentrations of serum BUN and Crea in adenine-induced CKD indicate kidney failure and the inability to filter waste effectively [49]. In CKD, the kidneys are unable to remove excess P and convert vitamin D to its active form, resulting in an imbalance of Ca and P in blood [50]. In this study, we found that treatment of adenine-induced CKD mice with HY7718 significantly reduced the abnormally high levels of these renal function markers and inorganic Ca and P in blood. In other words, HY7718 effectively alleviated the symptoms of CKD by suppressing increases in the levels of blood BUN, Crea and inorganic minerals via improvements in renal function.

Histopathological scoring showed that adenine diet supplementation significantly elevated tubular injury including tubular necrosis, cast formation, and tubular dilatation. Tubular necrosis is characterized by damage to the kidney’s tubules caused by the lack of blood flow or exposure to nephrotoxins. Injured or damaged tubular epithelial cells lead to the sloughing of cells and debris into the tubule lumen, resulting in the formation of casts containing aggregated tubule cells. The damaged tubules may also abnormally widen or enlarge, a process known as tubular dilatation [51,52]. In this study, we showed that APN or HY7718 treatment improved kidney histopathological score in an adenine-induced CKD model. Especially, oral intake of HY7718 was more effective than that of APN in alleviating kidney injury. Taken together, our data demonstrated that HY7718 may mitigate the progression of CKD by attenuating certain histopathological features.

The observation of an increase in the monocellular cell infiltration grade of adenine-induced CKD mice in this study confirmed that renal inflammation was an active feature of this CKD model. Monocellular cell infiltration is the accumulation of immune cells, such as T lymphocytes, monocytes, and macrophages, in kidney tissues. The infiltrating cells commonly contribute to kidney damage or immune responses [53]. Additionally, abnormal activation of the TLR4 and NF-κB signaling pathways leads to the production of pro-inflammatory molecules that accelerate the development/progression of CKD [54]. We confirmed the upregulation of inflammatory responses in the adenine-induced CKD mice by demonstrating increases in the expression of genes encoding pro-inflammatory cytokines and members of the TLR/NF-κB signaling pathway in the kidney. HY7718 treatment almost completely eliminated the monocellular cell infiltration of the adenine-induced CKD mice, and suppressed the expression of inflammatory cytokines and immune response-related genes.

Finally, we analyzed the effect of HY7718 on the apoptosis in the adenine-induced CKD mice by detecting tubule apoptotic cells, and measuring the *Bax/Bcl-2* ratio and *Casp3* gene expression. Excessive increases in apoptotic cells as well as the *Bax/Bcl-2* ratio and *Casp3* expression in the CKD mice was significantly attenuated by HY7718 treatment.

We believe that it is significant that HY7718 was more effective in alleviating the pathophysiological features of CKD than the positive control, APN. This may suggest that HY7718 may have therapeutic potential in alleviating CKD by improving kidney function and reducing tubular injury, inflammation, and apoptosis. Recent studies have reported that probiotics may improve CKD by leveraging bidirectional communication between kidney function and gut microbiota, known as the “gut-kidney axis” [10,55]. Further studies are required to examine the effects of orally administrated HY7718 on intestinal barrier integrity, gut microbial balance, and regulation of microbiota-derived metabolites.

## 4. Materials and Methods

### 4.1. Bacterial Strain Culture and Sample Preparation

*Lactiplantibacillus plantarum* HY7718 (HY7718), isolated from Korean traditional fermented foods, was provided hy Co., Ltd. (Yongin-si, Republic of Korea). HY7718 bacterial stocks were maintained in MRS broth (BD Difco, Sparks, MD, USA) with 20% (*v*/*v*) glycerol (Catalog number: G9012, Sigma-Aldrich, St. Louis, MO, USA) and were stored at −80 °C. HY7718 was cultured at 37 °C for 24 h in MRS broth under anaerobic conditions. For in vitro experiments, HY7718 cells were pelleted by centrifuging at 4000× *g* for 30 min. The resulting HY7718 pellet was washed twice and resuspended in sterile phosphate-buffered saline. For animal experiments, fresh cultured HY7718 was freeze-dried, and supplied as animal feed.

### 4.2. Cell Culture of HK2 Cells and Sample Treatment

The HK2 epithelial cell line, derived from normal human adult kidney, was purchased from the Korean Cell Line Bank (Seoul, Republic of Korea) and was maintained at 37 °C in a 5% CO_2_ incubator in RPMI 1640 Medium (Gibco, Waltham, MA, USA) containing 10% heat-inactivated FBS (Gibco, Waltham, MA, USA) and 1% penicillin/streptomycin (p/s, Gibco, Waltham, MA, USA).

For the oxidative stress-induced model, the HK2 cells were seeded in to 12-well plates (1 × 10^5^ cells/well) and grown for 24 h in 37 °C/5% CO_2_. After acclimation, the medium was replaced with RPMI 1640 without FBS and p/s to induce starvation conditions for 24 h. The cells were then pre-treated for 4 h with 10^6^ or 10^7^ CFU/mL of HY7718, after which 100 μM H_2_O_2_ (Sigma-Aldrich, St. Louis, MO, USA) was added for 4 h. The negative control wells were treated with only 100 μM H_2_O_2_.

For the apoptosis-induced model, the HK2 cells were seeded into 12-well plates (1 × 10^5^ cells/well) at 37 °C for 24 h. The growth medium was removed and replaced with RPMI 1640 without serum and antibiotics for 24 h. Next, 10^6^ or 10^7^ CFU/mL of HY7718 were pre-added into the wells and incubated for 4 h. Then, 5 ng/mL tumor necrosis factor-α (TNF, R&D Systems, Minneapolis, MN, USA) and 25 nM cycloheximide (CHX, Sigma-Aldrich, St. Louis, MO, USA) were added to the cells and incubated overnight. The negative control wells were treated with 5 ng/mL TNF and 25 nM CHX only.

### 4.3. Cell Viability Assay

HK2 was seeded into 96-well plates (2 × 10^4^ cells/well) and grown for 24 h. The cells were pre-treated for 24 h with 10^6^ or 10^7^ CFU/mL of HY7718 and then exposed to 100 μM H_2_O_2_ or 5 ng/mL TNF and 1 nM CHX for 18 h. After treatment, 10 μL of 0.5 mg/mL MTT [3-(4,5-dimethylthiazol-2-yl)-2,5-diphenyltetrazolium bromide] solution was added to each well, and the cells were incubated for a further 4 h until purple formazan crystals formed. After the medium containing MTT solution was removed, 100 μL dimethyl sulfoxide (DMSO) was added to dissolve the insoluble formazan crystals. The number of viable cells was quantified by measuring absorbance at 570 nm in a BioTek^®^ Synergy HT Microplate reader (Santa Clara, CA, USA).

### 4.4. Cytotoxicity Assay

Levels of LDH release was analyzed using the CytoTox 96 Non-Radioactive Cytotoxicity Assay Kit (Promega, Madison, WI, USA). Cytotoxicity was measured at an absorbance of 490 nm, and percent cytotoxicity was calculated as follows: 100 × (Experimental LDH release/Maximum LDH release).

### 4.5. Secretion of Pro-Inflammatory Cytokines

The secretion of TNF and IL-6 into HK2 cell culture medium was measured using the BD OptEIA™ Human TNF ELISA Set and BD OptEIA™ Human IL-6 ELISA Set (BD Biosciences, San Diego, CA, USA). Absorbance at 450 nm was measured in a BioTek^®^ Synergy HT Microplate reader.

### 4.6. Measurement of Caspase-3 Activity and Cleaved Caspase-3

Levels of caspase-3 activity and cleaved caspase-3 in the apoptosis-induced HK2 cells model were measured using the Caspase-3 Assay Kit (Colorimetric) (Abcam, Cambridge, UK) and Human Cleaved Caspase-3 (Asp175) ELISA Kit. Absorbance at 400 nm (caspase-3) or 450 nm (cleaved caspase-3) was measured in a BioTek^®^ Synergy HT Microplate reader.

### 4.7. Animal Experiments Design

For animal studies, a CKD model was established by feeding C57BL/6 mice with an 0.15% adenine-supplemented diet, a widely used method for generating CKD model mice [21]. Eight-week-old male C57BL/6 mice were purchased from Dooyeol Biotech (Seoul, Republic of Korea) and housed in cages with a humidity of 55 ± 10%, a temperature of 22 ± 1 °C, and a 12 h light/dark cycle. After 7 days of housing, the mice were randomly divided into four groups (*n* = 10/group): a control group (fed an AIN-93G diet (D10012G, Research Diets, New Brunswick, NJ, USA)); a CKD group (fed a 0.15% adenine-supplemented AIN-93G diet); an APN group (fed a 0.15% adenine-supplemented AIN-93G diet + allopurinol (APN, positive control, 20 mg/kg/day)); and an HY7718 group (fed a 0.15% adenine-supplemented AIN-93G diet + HY7718 (10^8^ CFU/kg/day).The APN and HY7718 were suspended in 200 μL of saline and administrated by oral gavage for animal experiments period. As controls, the control and CKD group were orally fed an equal amount of saline for the same period instead of APN or HY7718.4 weeks after starting the sample administration, the mice were fed a 0.15% adenine-supplemented diet to induce CKD During 3 weeks. Body weight, food intake, and water consumption were measured every week during experiments. After inducing CKD, the mice were sacrificed. Kidney and spleen tissue masses were measured immediately after sacrifice. Blood, urine, and kidney samples were collected for further analysis and stored at −80 °C. All experimental protocols were approved by the Ethics Review Committee of R&BD Center, hy Co., Ltd., Republic of Korea (AEC-2025-0003-Y). Figure 9 shows a flow chart of the animal experiments.

### 4.8. Serum Biochemical Analysis

Blood samples from mice were left at room temperature for 60 min and then centrifuged at 3000× *g* for 20 min at 4 °C to separate the serum. Levels of renal function markers (BUN and Crea) and inorganic Ca and P in serum samples were measured. All serum analyses were performed using a 7180 Clinical Analyzer (Hitachi, Tokyo, Japan) by Dooyeol Biotech (Seoul, Republic of Korea).

### 4.9. Histopathological Examination

Kidney tissues were excised and immediately fixed in 10% formalin solution. Fixed samples were embedded in paraffin, sectioned, and stained with H&E. Stained slides were photographed by MoticDSAssistant (Motic VM V1 Viewer 2.0). Histopathology was evaluated by Dooyeol Biotech. Kidney histopathological score was calculated as the sum of the indices of tubular necrosis, cast formation, and tubular dilatation in sections. Each index was scored on a scale of 0 to 5 for non-overlapping visual fields (×200) as follows: 0 (none), 1 (≤10%), 2 (11–25%), 3 (26–45%), 4 (46–75%) and 5 (≥75%). Mononuclear cell infiltration was also scored in the same manner.

### 4.10. Detection of Apoptosis by the TUNEL Assay

Apoptotic cells in kidney tissues were detected by the TUNEL assay with 3,3-diaminobenxidine (DAB). The stained slides were captured by MoticDSAssistant (Motic VM V1 Viewer 2.0). TUNEL-positive apoptotic cells (%) were measured by ImageJ v 1.54 software (https://imagej.net/ij/, accessed on 16 September 2025).

### 4.11. Quantitative Real-Time Polymerase Chain Reaction (PCR)

Total RNA was extracted from HK2 cells and kidney tissues using Easy-spin Total RNA Extraction Kit (iNtRON Biotechnology, Seoul, Republic of Korea). Eluted RNA was converted to cDNA at 37 °C for 1 h using the Omniscript Reverse Transcription Kit (Qiagen, Hilden, Germany), and the amount of cDNA was quantified. Quantitative real-time PCR was performed with TaqMan^TM^ Gene Expression Master Mix (Applied Biosystems, Waltham, MA, USA) on the QuantStudio 6 Flex Real-time PCR System (Applied Biosystems). The human and mouse gene-specific probes used for gene expression analysis are shown in Table 1. The results were determined as the relative expression normalized to that of a house-keeping gene (*GAPDH*, glyceraldehyde-3-phosphate dehydrogenase).

### 4.12. Statistical Analysis

The data were analyzed and graphed using GraphPad Prism 10 software (GraphPad Software, San Diego, CA, USA). Results are represented as mean ± standard deviation (SD). One-way ANOVA with Tukey’s post hoc test was applied to make comparisons between groups. Statistical significance was considered as *p* < 0.05.

## 5. Conclusions

Our results demonstrate that HY7718 ameliorates CKD in both in vitro and in vivo models. In oxidative stress- and apoptosis-induced HK2 cells, HY7718 suppressed pro-inflammatory cytokine secretion, reduced caspase-3 activity, and lowered cleaved caspase-3 levels. In adenine-induced CKD mice, oral administration of HY7718 improved renal function markers, including BUN and Crea, and alleviated histological kidney injury such as tubular necrosis, cast formation, and tubular dilatation. HY7718 also attenuated renal inflammation by reducing mononuclear cell infiltration and downregulating pro-inflammatory cytokine and immune-related gene expression, while exerting anti-apoptotic effects through reductions in tubular apoptotic cells and apoptosis-related genes. Taken together, these findings suggest that HY7718 may be a potential probiotic candidate for CKD management. Future studies will investigate its effects on intestinal barrier function and gut microbiota to clarify the involvement of the gut–kidney axis.

## Figures and Tables

**Figure 1 ijms-26-10052-f001:**
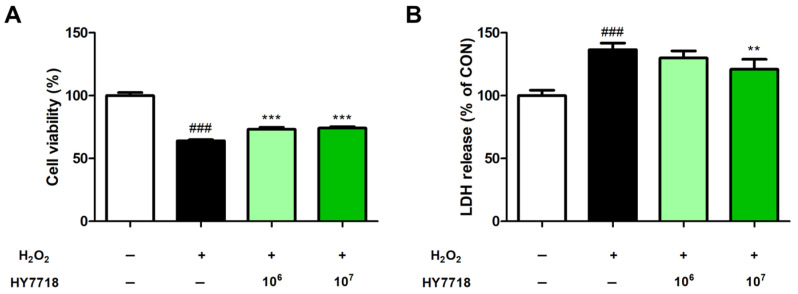
The effect of HY7718 on the viability of H_2_O_2_-treated HK2 cells. (**A**) Cell viability. (**B**) LDH release. Data are presented as the mean ± SD. ^###^
*p* < 0.001 vs. untreated cells. ** *p* < 0.01 and *** *p* < 0.001 vs. H_2_O_2_-treated cells. H_2_O_2_, hydrogen peroxide; LDH, lactate dehydrogenase; HY7718, pre-treatment with *Lactiplantibacillus plantarum* HY7718.

**Figure 2 ijms-26-10052-f002:**
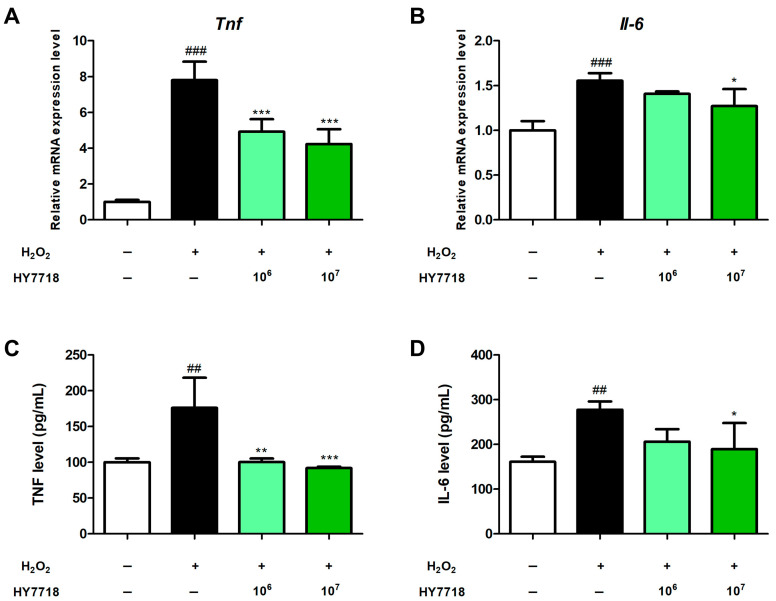
Effect of HY7718 on pro-inflammatory cytokine expression and secretion in H_2_O_2_-treated HK2 cells. Gene expression of (**A**) *Tnf* and (**B**) *Il-6*. Secretion level of (**C**) TNF and (**D**) IL-6. Data are presented as the mean ± SD. ^##^
*p* < 0.01 and ^###^
*p* < 0.001 vs. untreated cells. * *p* < 0.05, ** *p* < 0.01, and *** *p* < 0.001 vs. H_2_O_2_-treated cells. H_2_O_2_, hydrogen peroxide; TNF, tumor necrosis factor-alpha; IL-6, interleukin-6; HY7718, pre-treatment with *Lactiplantibacillus plantarum* HY7718.

**Figure 3 ijms-26-10052-f003:**
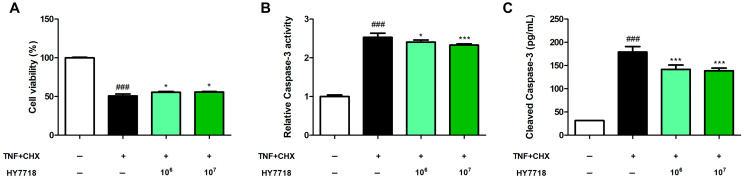
The effect of HY7718 on the cell viability, caspase-3 activity, and the cleaved caspase-3 level in TNF+CHX-treated HK2 cells. (**A**) Cell viability, (**B**) Relative Caspase-3 activity, (**C**) Levels of cleaved Caspase-3. Data are presented as the mean ± SD. ^###^
*p* < 0.001 vs. untreated cells. * *p* < 0.05 and *** *p* < 0.001 vs. TNF+CHX-treated cells. TNF, tumor necrosis factor-alpha; CHX, cyclohexamide; HY7718, pre-treatment with *Lactiplantibacillus plantarum* HY7718.

**Figure 4 ijms-26-10052-f004:**
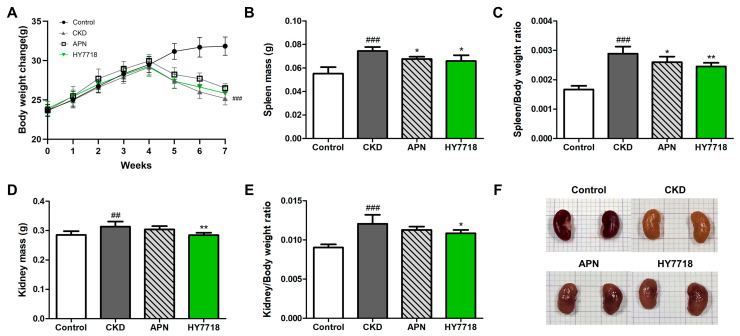
Effect of HY7718 on physiological parameters of adenine-induced CKD mice. (**A**) Change in body weight, (**B**) spleen mass, (**C**) spleen mass/body weight ratio, (**D**) kidney mass, (**E**) kidney mass/body weight ratio, and (**F**) morphology of kidney tissues. Data are presented as mean ± SD. ^##^
*p* < 0.01 and ^###^
*p* < 0.001 vs. control mice. * *p* < 0.05 and ** *p* < 0.01 vs. CKD group. CKD, adenine-fed mice; APN; allopurinol (20 mg/kg/day) + adenine; HY7718, *Lactiplantibacillus plantarum* HY7718 (10^8^ CFU/kg/day) + adenine.

**Figure 5 ijms-26-10052-f005:**
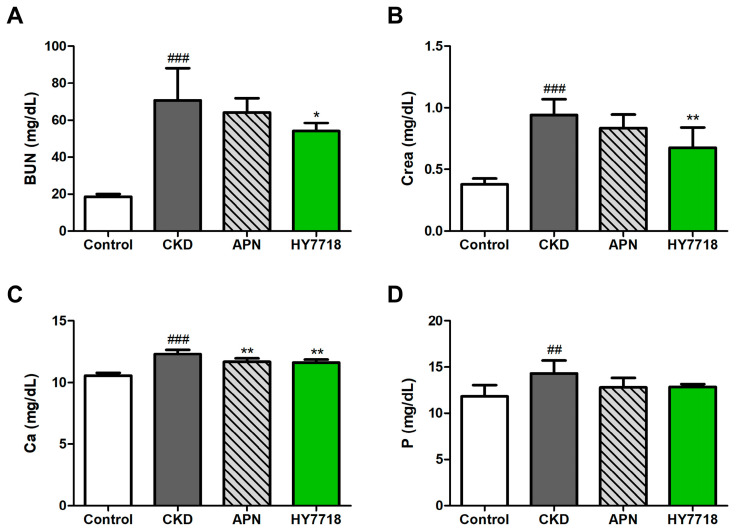
Effect of HY7718 and allopurinol on levels of blood biochemical indicators in adenine-induced CKD mice. Concentration of (**A**) BUN, (**B**) Crea, (**C**) Ca, (**D**) P. Data are presented as mean ± SD. ^##^
*p* < 0.01 and ^###^
*p* < 0.001 vs. control mice. * *p* < 0.05 and ** *p* < 0.01 vs. CKD mice. CKD, adenine-induced CKD mice; APN; allopurinol (20 mg/kg/day) + adenine; HY7718, *Lactiplantibacillus plantarum* HY7718 (10^8^ CFU/kg/day) + adenine; BUN, blood urea nitrogen; Crea, creatinine; Ca, calcium; P, phosphorus.

**Figure 6 ijms-26-10052-f006:**
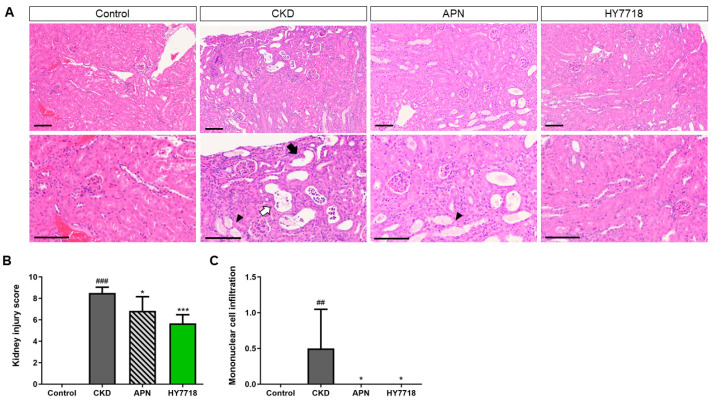
Kidney histopathological analysis of adenine-induced CKD mice. (**A**) Images of hematoxylin and eosin-stained kidney tissue sections (×200 and ×400 magnification; Scale bar = 100 μm), (**B**) Kidney histopathological score and (**C**) Mononuclear cell infiltration score. Black arrow, tubular dilatation; white arrow, tubular necrosis; black arrow head, cast formation. Data are presented as mean ± SD. ^##^
*p* < 0.01 and ^###^
*p* < 0.001 vs. control mice. * *p* < 0.05 and *** *p* < 0.001 vs. CKD mice. CKD, adenine-induced CKD mice; APN, allopurinol (20 mg/kg/day) + adenine; HY7718, *Lactiplantibacillus plantarum* HY7718 (10^8^ CFU/kg/day) + adenine.

**Figure 7 ijms-26-10052-f007:**
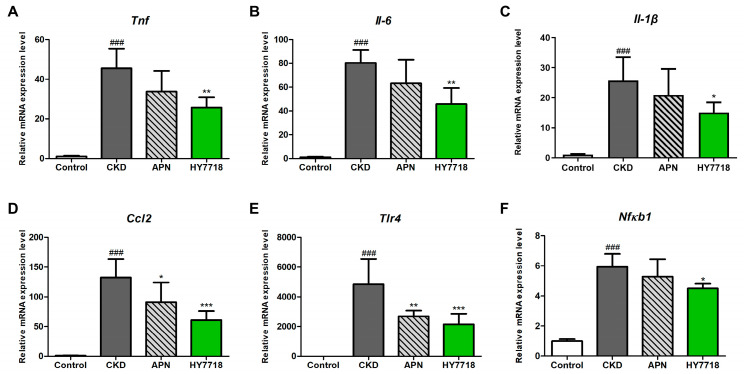
mRNA expression of inflammation-related genes in adenine-induced CKD mice with and without HY7718 or APN treatment. (**A**) *Tnf*, (**B**) *Il-6*, (**C**) *Il-1β*, (**D**) *Ccl2* (**E**) *Tlr4*, and (**F**) *Nfκb1*. Data are presented as the mean ± SD. ^###^
*p* < 0.001 vs. control mice. * *p* < 0.05, ** *p* < 0.01 and *** *p* < 0.001 vs. CKD mice. CKD, adenine-induced CKD mice; APN, allopurinol (20 mg/kg/day) + adenine; HY7718, *Lactiplantibacillus plantarum* HY7718 (10^8^ CFU/kg/day) + adenine; *Tnf*, tumor necrosis factor-alpha; *Il-6*, interleukine-6; *Il-1β*, Interleukin 1 beta; *Ccl2*, C-C motif chemokine ligand 2; *Tlr4*, Toll-like receptor 4; *Nfκb1*, nuclear factor kappa B subunit 1.

**Figure 8 ijms-26-10052-f008:**
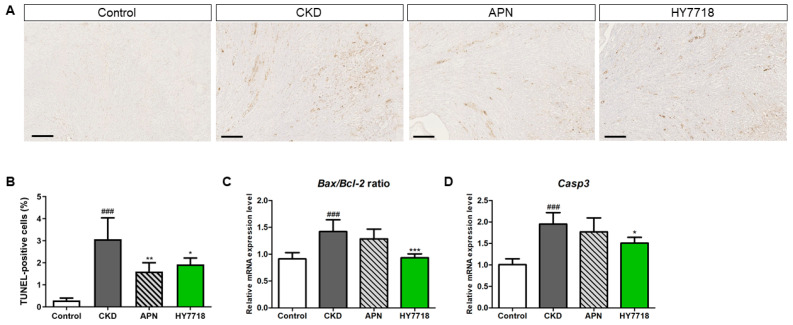
Apoptosis in adenine-induced CKD mice in the presence and absence of HY7718 or APN. (**A**) Apoptotic cells revealed by TUNEL staining (Scale bar = 300 μm), (**B**) TUNEL-positive cells (%), (**C**) *Bax/Bcl-2* expression ratio, and (**D**) *Casp3* expression. Data are presented as the mean ± SD. ^###^
*p* < 0.001 vs. control mice. * *p* < 0.05, ** *p* < 0.01 and *** *p* < 0.001 vs. CKD mice. CKD, adenine-induced CKD mice; APN, allopurinol (20 mg/kg/day) + adenine; HY7718, *Lactiplantibacillus plantarum* HY7718 (10^8^ CFU/kg/day) + adenine; TUNEL, terminal deoxynucleotidyl transferase dUTP nick end labeling; *Bax*, BCL2 associated X; *Bcl-2*, B cell lymphoma 2; *Casp3*, Caspase-3.

**Figure 9 ijms-26-10052-f009:**
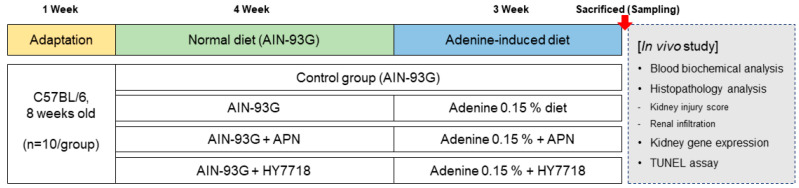
Flow chart of animal experiments.

**Table 1 ijms-26-10052-t001:** Information on the Taqman probes used in vitro and in vivo experiments.

Gene Category	Gene Symbol	Gene Name	Assay ID
In vitro experiments (HK cells)			
House-keeping gene	*GAPDH*	Glyceraldehyde-3-phosphate dehydrogenase	Hs99999905_m1
Pro-inflammatory cytokines	*TNFα*	Tumor necrosis factor-alpha	Hs00174128_m1
	*IL-6*	Interleukin-6	Hs00174131_m1
In vivo experiments (Mouse kidney tissues)			
House-keeping gene	*Gapdh*	Glyceraldehyde-3-phosphate dehydrogenase	Mm99999915_g1
Pro-inflammatory cytokines	*Tnfα*	Tumor necrosis factor-alpha	Mm00443258_m1
	*Il-6*	Interleukin-6	Mm00446190_m1
	*Il-1* *β*	Interleukin 1 beta	Mm00434228_m1
	*Ccl2* (*Mcp-1*)	C-C motif chemokine ligand 2	Mm00441242_m1
Inflammatory signaling pathway	*Tlr4*	Toll-like receptor 4	Mm00445273_m1
	*Nfκb1*	Nuclear factor kappa B subunit1	Mm00476361_m1
Apoptosis	*Bax*	BCL2 associated X, apoptosis regulator	Mm00432051_m1
	*Bcl-2*	BCL2, apoptosis regulator	Mm00477631_m1
	*Casp3*	Caspase-3	Mm01195085_m1

## Data Availability

The original contributions presented in this study are included in the article/Appendix A. Further inquiries can be directed to the corresponding author.

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
