# Peer review of "Lactiplantibacillus plantarum* HY7718 Attenuates Renal Injury in an Adenine-Induced Chronic Kidney Disease Mouse Model via Inhibition of Inflammation and Apoptosis"

_ijms, 2025, doi:10.3390/ijms262010052_

Round 1

Reviewer 1 Report

Comments and Suggestions for Authors

Report is attached

Reviewer 2 Report

Comments and Suggestions for Authors

The manuscript presents a well-structured set of in vitro and in vivo experiments, convincingly demonstrating the potential benefits of treatment with Lactiplantibacillus plantarum HY7718 in an oxidative stress model using the HK-2 cell line and in an adenine-induced chronic kidney disease (CKD) model in mice.

I recommend the manuscript for publication after minor revisions, with particular attention to the Materials and Methods section to improve clarity:

  • Line 356 – “as animal feed” - Please clarify precisely what is meant by “as animal feed,” especially relative to the in vitro
  • Line 352: Please provide the catalog number of glycerol and describe the thawing temperature regimen. These are critical for preserving bacterial viability after long-term storage.
  • Line 363 – Please specify the cell seeding density and the duration of the serum-starvation period.
  • Line 369 - As above, please include the seeding density and starvation time.
  • Lines 372–375 – Please cite the source(s) for the selected concentrations (TNF-α, cycloheximide) or briefly summarize your dose–response data supporting these choices.
  • Line 409 – Please provide either as supplementary material the full composition of the AIN-93G diet (or provide a citation to the complete formula).
  • Line 404 – Provide a brief justification for the chosen adenine percentage and cite supporting literature.
  • Line 410 – Clarify whether the APN diet has the same base composition as the control diet and define APN in full at first mention.
  • Lines 402–415 - the text is difficult to follow. I recommend rewriting this section to provide a more precise explanation of all experimental groups, diet compositions, and exposure durations.
  • Line 425 – Please, indicate how long blood samples were allowed to clot before centrifugation.
  • Several abbreviations are not defined (e.g., 2,8-DHA, BUN, Crea, APN). Please explain all abbreviations at first mention or provide an abbreviations list.
  • In the PDF I reviewed, some figures appear slightly blurred (axis labels, abbreviations, significance markers). Please ensure that high-resolution figures (at least 300 dpi) are used.
  • Lines 156–158 – Please mention that the commented results are not statistically significant to avoid a speculative tone.

Best regards,
